# Calculation of the Cosmological Constant for the Planetary System in Schwarzschild's Cosmological Model

**Alvaro Humberto Salas Salas [1], Jairo Ernesto Castillo Hernandez [2],[*] and Jorge Enrique Pinzon Quintero [3]**

[1] FIZMAKO Research Group, Department of Mathematics and Statistics, Universidad Nacional de Colombia Sede Manizales, Manizales 500001, Colombia
[2] FIZMAKO Research Group, Universidad Distrital Francisco José de Caldas, Bogotá 11021, Colombia
[3] FIZMAKO Research Group, Universidad del Tolima, Ibagué 730001, Colombia
[*] Correspondence: jcastillo@udistrital.edu.co

**Abstract:** In this work, the static cosmological model of the Schwarzschild solution for the solar system is proposed taking into account the cosmological constant in the equation of the general theory of relativity (GTR) proposed by A. Einstein. We found the nonlinear differential equation that describes the behavior of the planets around the Sun; this is solved exactly by the Jacobi and Weierstrass elliptic functions. The obtained solution allows for us to estimate the value of the cosmological constant knowing the perihelion of the different planets and from different mathematical approaches; that is, the inverse problem is solved. From the obtained results, the Schwarzschild static cosmological model for the solar system is proposed, establishing the Schwarzschild cosmological radius and the curvature limit of the solar system. From the curvature limit, different regions are proposed for the planets, exoplanets, and a region is predicted where the existence of new planets and exoplanets belonging to the solar system is possible. The proposed theory of the static Schwarzschild cosmological model may be of great interest to astronomers, cosmologists, and all those interested in the study of the universe.

**Keywords:** cosmological model; Schwarzschild model; static cosmological model

## 1. Introduction

Albert Einstein in 1915 [1,2] formulated the general theory of gravitation (TGR), synthesizing it in the equation

$$R_{\alpha\beta} - \frac{1}{2} R g_{\alpha\beta} = \frac{8\pi G}{c^4} T_{\alpha\beta} \tag{1}$$

He verified that his theory explained the anomaly of the perihelion of Mercury and the curvature of light in the gravitational field of the Sun. The first exact static solution was found by Schwarzschild [3]. This solution led to deducing two great physical phenomena in our planetary system: the perihelion of the planet Mercury, the deviation of light rays in the gravitational field of the Sun, establishing that spacetime in the vicinity of a gravitational mass is curved, later experimentally demonstrated [4,5], this being a great triumph for GTG.

In the following years and recently, nonstatic solutions were found, such as those of: E. Kasner [6–9], H.P Robertson [10], A.G Walker [11], A. Friedman [12], and G. Lemaitre [13], which are called the Friedmann–Lemaitre–Robertson–Walker metric or FLRW cosmological model.

In 1917, Albert Einstein [14] could not find a static universe of his original equation. This led him to reformulate the original equation of the GTG, introducing a term called the lambda Λ-cosmological constant, which encompass the physical structure of the static universe. Later, in 1937, Albert Einstein rejected this idea once De sitter [15] had established that there is a solution of Equation (1) for empty space, which represents a model of the

expanding universe, and it was observed by Edwin Hubbles [16] in the redshift of galaxies, suggesting that the universe was not static, and that Eddington [17] demonstrated the expansion of the universe and that the static universe with cosmological constant was unstable [18,19].

$$R_{\alpha\beta} - \frac{1}{2}Rg_{\alpha\beta} - \Lambda g_{\alpha\beta} = \frac{8\pi G}{c^4}T_{\alpha\beta} \tag{2}$$

Edwin Hubbles' discovery of the redshift, and subsequent observations of different galaxies and supernovae yielded new results that confirmed the theory of redshift, the discoveries of the radiation of the cosmic background, the acceleration of the universe, the large-scale structure of the universe, dark energy, and dark matter, leading to retaking the importance of the cosmological constant and the search for solutions to the equations proposed by A. Einstein with the cosmological constant.

Many works studied the perihelion of Mercury [20,21] and of other planets with constant zero and nonzero cosmology [22–25] .

This work is organized as follows: In Section 1, Einstein's equation is solved with the cosmological constant in vacuum with the aim that the work is complete and independent for the reader. In Section 2 are the algebraic equations of geodesics, and the nonlinear differential equation that describes the behavior of a planet around the Sun. In Sections 3 and 4, the nonlinear differential equation is solved by means of the elliptic Jacobi functions and Weierstrass elliptic function. The cosmological constant is found as a function of the perihelion and aphelion of the planets around the Sun from different perspectives. Lastly, an exhaustive analysis of the obtained results and their consequences in Schwarzschild's cosmological static model was carried out, for which A. Einstein so yearned and sought.

## 2. Solution of Einstein's Equation with the Cosmological Constant

In this section, we propose to derive the Schwarzschild metric while taking into account the cosmological constant. For this purpose, we solve Einstein's equation in a vacuum while taking into account the cosmological constant

$$R_{\alpha\beta} - \Lambda g_{\alpha\beta} = 0 \tag{3}$$

where $R_{\alpha\beta}$ is the Ricci tensor, $g_{\alpha\beta}$ the metric tensor, and $\Lambda$ the cosmological constant.

Since the Schwarzschild solution is static and isotropic, let us find the solution of (3) of the form:

$$ds^2 = D(r)dt^2 - E(r)dr^2 - F(r)r^2(d\theta^2 + \sin^2\theta d\phi^2) \tag{4}$$

Now, we perform a coordinate transformation $(ct, r, \theta, \phi) -> (ct, \rho, \theta, \phi)$ where the new coordinate $\rho$ is defined by the equation
$\rho^2 \equiv F(r)r^2$:

$$ds^2 = A(\rho)c^2dt^2 - B(\rho)d\rho^2 - \rho^2(d\theta^2 + \sin^2\theta d\phi^2) \tag{5}$$

The metric tensor that corresponds to the linear differential element is clearly

$$g_{\alpha\beta} = \begin{pmatrix} A(\rho) & 0 & 0 & 0 \\ 0 & -B(\rho) & 0 & 0 \\ 0 & 0 & -\rho^2 & 0 \\ 0 & 0 & 0 & -\rho^2\sin^2\theta \end{pmatrix} \tag{6}$$

This is the metric of the more general isotropic static field problem. Functions $A(\rho)$ and $B(\rho)$ are still indeterminate, as is the relationship between coordinates $\rho$ and $r$. When calculating the components of Ricci tensor $R_{\alpha\beta} = \partial_\rho \Gamma^\rho_{\beta\alpha} - \partial_\beta \Gamma^\rho_{\rho\alpha} + \Gamma^\rho_{\rho\lambda}\Gamma^\lambda_{\beta\alpha} - \Gamma^\rho_{\beta\lambda}\Gamma^\lambda_{\rho\alpha}$, the only nonzero components are:

$$R_{00} = \frac{A(\rho)(2B(\rho)(\rho A''(\rho)+2A'(\rho))-\rho A'(\rho)B'(\rho))-\rho B(\rho)A'(\rho)^2}{4\rho A(\rho)B(\rho)^2}$$
$$R_{11} = \frac{A(\rho)(\rho A'(\rho)+4A(\rho))B'(\rho)+\rho B(\rho)\left(A'(\rho)^2-2A(\rho)A''(\rho)\right)}{4\rho A(\rho)^2 B(\rho)}$$
$$R_{22} = \frac{-\rho B(\rho)A'(\rho)+\rho A(\rho)B'(\rho)^2-2A(\rho)B(\rho)}{2A(\rho)B(\rho)^2}$$
$$R_{33} = \frac{\sin^2(\phi)\left(A(\rho)\left(\rho B'(\rho)^2-2B(\rho)\right)-\rho B(\rho)A'(\rho)\right)}{2A(\rho)B(\rho)^2}$$

(7)

Replacing (7) and (6) in (3), we obtain the following system of equations for $A(\rho)$ end $B(\rho)$:

$$2\rho A(\rho)B(\rho)A''(\rho) - \rho A(\rho)A'(\rho)B'(\rho) - \rho B(\rho)A'(\rho)^2 + 4A(\rho)B(\rho)A'(\rho)^2 B(\rho)^2 + \Lambda A(\rho) = 0$$
$$-2\rho A(\rho)B(\rho)A''(\rho) + \rho A(\rho)A'(\rho)B'(\rho) + \rho B(\rho)A'(\rho)^2 + 4A(\rho)^2 B'^2 B(\rho)^2 - \Lambda B(\rho) = 0$$
$$-\rho B(\rho)A'(\rho) + \rho A(\rho)B'(\rho)^2 A(\rho)B(\rho)^2 + 2A(\rho)B(\rho)^2 - 2A(\rho)B(\rho) - \Lambda \rho^2 = 0$$
$$-\sin^2(\phi)\left(\rho B(\rho)A'(\rho) - \rho A(\rho)B'(\rho)^2 A(\rho)B(\rho)^2 - 2A(\rho)B(\rho)^2 + 2A(\rho)B(\rho)\right) - \Lambda \rho^2 \sin^2(\theta) = 0$$

(8)

Adding the first and second equations of System (8), we have $A(\rho)B'(\rho) + A'(\rho)B(\rho) = (AB)' = 0$. This means that:

$$A(\rho)B(\rho) = k_1 , \tag{9}$$

so that $B(\rho) = \frac{k_1}{A(\rho)}$. By placing this equation into the third equation of System (8), we obtain:

$$\rho A'(\rho) + A(\rho) + k_1 \Lambda \rho^2 - k_1 = 0, \tag{10}$$

from where:

$$A(\rho) = k_1 - \frac{1}{3}k_1 \rho^2 \Lambda + \frac{c_1}{\rho} \tag{11}$$

In order to determine the values of constants $k_1$ and $c_1$, some physical condition must be used, which is achieved by studying the situation at the Newtonian limit. When $\Lambda = 0$, the Newtonian potential is $\Phi = -\frac{GM}{r}$; therefore,

$$g_{00} = A(\rho) \simeq 1 - \frac{2GM}{c^2 r} \tag{12}$$

Comparing with (11), we see that $k_1 = 1$ and $\frac{c_1}{\rho} = -\frac{2GM}{c^2 r}$　The set of two variables $(c_1, \rho)$ is controlled by a single equation. This means that the values of variables $(c_1, \rho)$ are undetermined. We then need to arbitrarily determine one of the variables. The most comfortable choice is $\rho = r$; consequently, $c_1$ is equal to $-\frac{2GM}{c^2}$. Therefore, we lastly have:

$$A = \frac{1}{B} = 1 - \frac{1}{3}r^2 \Lambda - \frac{2GM}{c^2 r} \tag{13}$$

Linear Differential Element (4) reads [15,26].

$$ds^2 = (1 - \frac{1}{3}r^2 \Lambda - \frac{2GM}{c^2 r})c^2 dt^2 - (1 - \frac{1}{3}r^2 \Lambda - \frac{2GM}{c^2 r})^{-1} dr^2 - r^2(d\theta^2 + \sin^2\theta d\phi^2), \tag{14}$$

where

$$g_{\alpha\beta} = \begin{pmatrix} (1 - \frac{1}{3}r^2\Lambda - \frac{2GM}{c^2 r}) & 0 & 0 & 0 \\ 0 & -(1 - \frac{1}{3}r^2\Lambda - \frac{2GM}{c^2 r})^{-1} & 0 & 0 \\ 0 & 0 & -r^2 & 0 \\ 0 & 0 & 0 & -r^2 \sin^2\theta \end{pmatrix} \tag{15}$$

## 3. Geodesic Equations and Equation of Motion

Let us consider the motion of a planet in the gravitational field of a much heavier body (the Sun). The gravitational field of the center in spherical coordinates is given by Linear Differential Element (14).

$$ds^2 = (1 - \frac{1}{3}r^2\Lambda - \frac{s}{r})c^2dt^2 - (1 - \frac{1}{3}r^2\Lambda - \frac{s}{r})^{-1}dr^2 - r^2(d\theta^2 + \sin^2\theta d\phi^2), \quad (16)$$

where $s = \frac{2GM}{c^2}$. With Metric (16), we calculate the Christoffel symbols.

$$\Gamma^\lambda_{\mu\nu} = \frac{1}{2}g^{\lambda\sigma}(\partial_\mu g_{\sigma\nu} + \partial_\nu g_{\sigma\mu} - \partial_\sigma g_{\mu\nu}) \quad (17)$$

The nonzero quantities are:

$$\begin{array}{lll}
\Gamma^0_{01} = \Gamma^0_{10} = \frac{3s - 2r^3\Lambda}{-2\Lambda r^4 + 6r^2 - 6sr} & \Gamma^1_{00} = \frac{c^2(\Lambda r^3 - 3r + 3s)(2r^3\Lambda - 3s)}{18r^3} & \Gamma^1_{11} = -\Gamma^0_{01} \\
\Gamma^1_{22} = \frac{r^3\Lambda}{3} - r + s & \Gamma^1_{33} = \frac{1}{3}(r^3\Lambda - 3r + 3s)\sin^2\theta & \Gamma^2_{12} = \Gamma^2_{21} = \frac{1}{r} \\
\Gamma^3_{13} = \Gamma^3_{31} = \frac{1}{r} & \Gamma^2_{33} = -\cos\theta\sin\theta & \Gamma^3_{32} = \Gamma^3_{23} = \cot\theta
\end{array} \quad (18)$$

Using the equations of geodesic lines $\frac{d^2x^\sigma}{d\tau^2} + \Gamma^\sigma_{\mu\nu}\frac{dx^\mu}{d\tau}\frac{dx^\nu}{d\tau} = 0$ and taking into account the central symmetry of our problem, any plane through the center can, however, be chosen as the plane $\theta = \frac{\pi}{2}$, that is, the orbit can be located in any plane through the center; we obtain the algebraic equations from the geodesics:

$$\frac{2r't'(3s - 2\Lambda r^3)}{-2\Lambda r^4 + 6r^2 - 6rs} + t'' = 0.$$
$$\frac{c^2(t')^2(\Lambda r^3 - 3r + 3s)(2\Lambda r^3 - 3s)}{18r^3} + \phi'^2\left(\frac{\Lambda r^3}{3} - r + s\right) + r''(k) - \frac{(r')^2(3s - 2\Lambda r^3)}{-2\Lambda r^4 + 6r^2 - 6rs} = 0. \quad (19)$$
$$\frac{2r'\phi'}{r} + \phi'' = 0.$$

From the first and third equations of (19), we obtain that:

$$t' = \frac{Er}{3s - 3r + r^3\Lambda}, \quad \phi' = \frac{L}{r^2}, \quad (20)$$

where $E$ and $L$ are the constants of integration that represent the effective energy of the system and the angular momentum.

Dividing (14) by $ds^2 = c^2d\tau^2$ gives

$$1 = -\frac{r'^2}{c^2\left(-\frac{1}{3}\Lambda r(\tau)^2 - \frac{s}{r(\tau)} + 1\right)} - \frac{r(\tau)^2\phi'^2}{c^2} + t'^2\left(-\frac{1}{3}\Lambda r(\tau)^2 - \frac{s}{r(\tau)} + 1\right) \quad (21)$$

Replacing (20) into (21):

$$-\frac{L^2}{c^2r(\tau)^2} - \frac{r'^2}{c^2\left(-\frac{1}{3}\Lambda r(\tau)^2 - \frac{s}{r(\tau)} + 1\right)} + \frac{E^2r(\tau)^2\left(-\frac{1}{3}\Lambda r(\tau)^2 - \frac{s}{r(\tau)} + 1\right)}{(\Lambda r(\tau)^3 - 3r(\tau) + 3s)^2} - 1 = 0 \quad (22)$$

Applying chain rule $r'(\tau) = \frac{dr}{d\varphi}\frac{d\varphi}{d\tau}$, $r'(\tau)^2 = (\frac{dr}{d\varphi})^2\frac{L^2}{r^4}$,

$$-r(\phi)^4\left(c^2\left(E^2 - 9\right) + 3\Lambda L^2\right) - 3c^2\Lambda r(\phi)^6 - 9c^2sr(\phi)^3 + 9L^2r'^2 - 9L^2sr(\phi) + 9L^2r(\phi)^2 = 0 \quad (23)$$

Using relation $r(\varphi) = \frac{1}{u(\varphi)}$ and taking into account that $s = \frac{2GM}{c^2}$, we obtain that

$$-\frac{c^2E^2}{9L^2} - \frac{2GMu(\phi)^3}{c^2} - \frac{c^2\Lambda}{3L^2u(\phi)^2} + \frac{c^2}{L^2} - \frac{2GMu(\phi)}{L^2} - \frac{\Lambda}{3} + u'^2 + u(\phi)^2 = 0 \quad (24)$$

or

$$-\frac{2GMu(\phi)^3}{c^2} - \frac{c^2\Lambda}{3L^2u(\phi)^2} - \frac{2GMu(\phi)}{L^2} + u'^2 + u(\phi)^2 = \frac{c^2E^2}{9L^2} - \frac{c^2}{L^2} + \frac{\Lambda}{3} = C \quad (25)$$

Derivating Equation (24) after simplifying (24), we obtain the equation of motion taking into account the cosmological constant:

$$\frac{d^2 u}{d\phi^2} + u - \frac{GM}{L^2} - \frac{3GM}{c^2} u^2 + \frac{c^2 \Lambda}{3L^2 u^3} = 0 \tag{26}$$

## 4. Einstein Equation for Planetary Motion

Let us consider the equation

$$u''(\phi) = \frac{GM}{L^2} - u(\phi) + \frac{3GM}{c^2} u^2(\phi), \ u(0) = \frac{1}{P} \text{ and } u'(0) = 0. \tag{27}$$

If we take into account cosmological constant $\Lambda$, the equation is modified by adding another term as follows:

$$u''(\phi) = \frac{GM}{L^2} - \frac{c^2 \Lambda}{3L^2 u^3(\phi)} - u(\phi) + \frac{3GM}{c^2} u^2(\phi) = 0, \ u(0) = \frac{1}{P} \text{ and } u'(0) = 0. \tag{28}$$

Equations (27) and (28) may both be solved in closed form. However, Equation (28) demands inverting some hyper elliptic Abelian integral and its solution is expressed in terms of a generalized Weierstrass function, which is a difficult task. Let us solve the easier Equation (27). Let

$$u(\phi) = \frac{1}{P} - S + S \operatorname{cn}^2 \left( \sqrt{\lambda + \mu} \phi, \frac{\mu}{2(\lambda + \mu)} \right). \tag{29}$$

Define the residual function

$$R(\phi) = \alpha - u(\phi) + \gamma u^2(\phi) - u''(\phi),$$

$$\alpha = \frac{GM}{L^2} = \frac{3AP(A+P) - 2\gamma\left(A^2 + AP + P^2\right)}{6A^2 P^2}, \gamma = \frac{3GM}{c^2}. \tag{30}$$

In these formulas, $A$ stands for the aphelion and $P$ stands for the perihelion of the planet. Number $L$ corresponds to the angular momentum of the planet. We have

$$R(\phi) = \frac{-P + P^2 S + P^2 \alpha + \gamma - 2PS\gamma + P^2 S^2 \gamma - 2P^2 S\lambda - P^2 S\mu}{P^2} -$$

$$\frac{(S(P - 2\gamma + 2PS\gamma - 4P\lambda))}{P} \operatorname{cn}^2 + S(S\gamma + 3\mu)\operatorname{cn}^4, \tag{31}$$

$$\operatorname{cn} = \operatorname{cn}\left( \sqrt{\lambda + \mu} \phi, \frac{\mu}{2(\lambda + \mu)} \right).$$

Equating the coefficients of $\operatorname{cn}^0$, $\operatorname{cn}^2$, and $\operatorname{cn}^4$ to zero gives an algebraic system whose solution is easily obtained, as follows:

$$\lambda = \frac{-2\gamma + 2\gamma PS + P}{4P}, \ \mu = -\frac{\gamma S}{3}.$$

$$S = \frac{6\gamma - 3P + \sqrt{-12\gamma^2 + 9P^2 + 12\gamma P(1 - 4\alpha P)}}{4\gamma P}. \tag{32}$$

Then,

$$u(\phi) = \frac{1}{P} - S + S \operatorname{cn}^2(\omega\phi, m), \tag{33}$$

where

$$m = \frac{1}{2} - \frac{3(\rho + 2\gamma - P)}{2(\rho - 6\gamma + 3P)} = \frac{2GM(A - P)}{2GM(2A + P) - Ac^2 P}. \tag{34}$$

$$\omega = \frac{1}{2}\sqrt{\frac{P - 2\gamma}{P(1 + m)}} \text{ and } S = \frac{1}{A} - \frac{3}{2\gamma} + \frac{2}{P}.$$

The perihelion shift is given by

$$\Delta_{GRT} = \left( \frac{2K(m)}{\omega} - 2\pi \right) \cdot \frac{23668612128}{\pi \cdot \text{Siederal}} \text{ arc-sec/cy.} \tag{35}$$

Here, Siederal stands for the orbit period of the planet. Since, for small $m$,

$$\frac{4K(m)}{\sqrt{\frac{P-2\gamma}{(m+1)P}}} - 2\pi \approx \frac{2\pi\left( 3m\sqrt{\frac{P-2\gamma}{P}} + 9m - 16\sqrt{\frac{P-2\gamma}{P}} + 16 \right)}{(16 - 3m)\sqrt{\frac{P-2\gamma}{P}}} \tag{36}$$

number $\Delta_{GRT}$ may be approximated as follows:

$$\tilde{\Delta}_{GRT} = \frac{47337224256\left( 8Ac^2P(1-v) + GM(29Av - 41A + 19Pv - 7P) \right)}{v(8Ac^2P - GM(29A + 19P)) \cdot \text{Siederal}} \text{ arcsec/cy,} \tag{37}$$
$$\text{where } v = \sqrt{1 - \frac{2\gamma}{P}} = \sqrt{1 - \frac{6GM}{c^2P}}$$

Another formula:

$$\hat{\Delta}_{GTR} = \frac{71005836384\mu\left( c^2P(Q+1) + 2\mu Q(Q+5) \right)}{c^4P^2\text{Siederal}} \text{ arcsec/century, } \mu = GM, \, Q = \frac{P}{A}. \tag{38}$$

*Exact Value for the Perihelion Shifts*

We used the following data:
Sun's mass: $M = 1.98854692413871773971206981288133967850592762187789 \times 10^{30}$
Gravitational constant: $G = 6.67383999879761360272969834 \times 10^{-11}$.
Speed of light in vacuum: $c = 299792458$
The results are depicted in Table 1.

**Table 1.** Exact and approximate values for perihelion shifts.

| Planet | $\Delta_{GTR}$ (Exact Value) | $\hat{\Delta}_{GTR}$ | $\lvert\Delta_{GTR} - \tilde{\Delta}_{GTR}\rvert$ |
|---|---|---|---|
| Mercury | 42.981518799592365 | 42.98151983984209 | $1.04 \times 10^{-6}$ |
| Venus | 8.625078587720143 | 8.625078762663101 | $1.75 \times 10^{-7}$ |
| Earth | 3.8387718031722122 | 3.8387718587113353 | $5.55 \times 10^{-8}$ |
| Mars | 1.3508767421610324 | 1.3508767534274582 | $1.13 \times 10^{-8}$ |
| Jupiter | 0.062311625478272306 | 0.06231162564333644 | $1.65 \times 10^{-10}$ |
| Saturn | 0.013688932210705418 | 0.013688932230249766 | $1.95 \times 10^{-11}$ |
| Uranus | 0.0023848021538887348 | 0.002384802155606265 | $1.72 \times 10^{-12}$ |
| Neptune | 0.000774085957956637 | 0.0007740859583331821 | $3.77 \times 10^{-13}$ |
| Pluto | 0.0004175446900080054 | 0.0004175446900902417 | $8.22 \times 10^{-14}$ |

Using NASA data, we obtained the following values for perihelion shifts:
Figure 1 illustrates the way in which Mercury's perihelion moves.

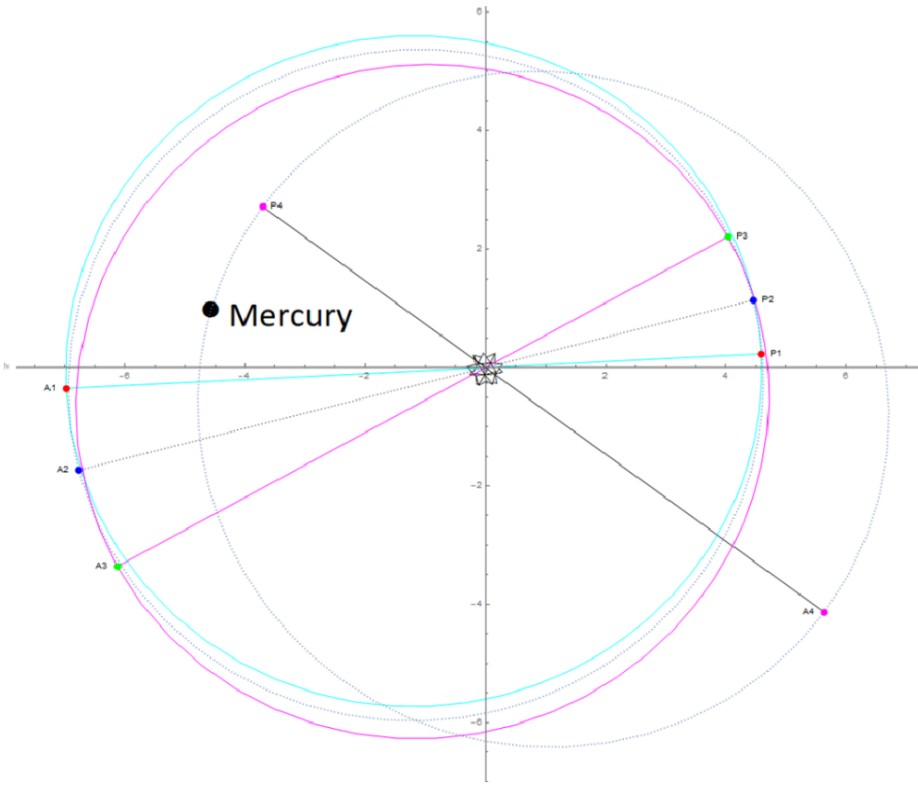

**Figure 1.** Advance of Mercury's perihelion.

**Remark 1.** *Equation (27) may also be written as*

$$\frac{du}{d\phi} = \sqrt{\frac{2\gamma}{3}} \sqrt{\left(u(\phi) - \frac{1}{A}\right)\left(\frac{1}{P} - u(\phi)\right)\left(\frac{3}{2\gamma} - \frac{1}{A} - \frac{1}{P} - u(\phi)\right)}. \tag{39}$$

*Integrating it gives*

$$T = 2(\phi(1/P) - \phi(1/A)) = \frac{4K\left(\frac{2GM(A-P)}{2GM(2A+P)-Ac^2P}\right)}{\sqrt{\frac{P-2\gamma}{P\left(1+\frac{2GM(A-P)}{2GM(2A+P)-Ac^2P}\right)}}} = \frac{2\sqrt{6A}K\left(\frac{A-P}{P(A\delta_0-1)}\right)}{\sqrt{\gamma}\sqrt{A\delta_0-1}}, \tag{40}$$

$$\delta_0 = \frac{3}{2\gamma} - \frac{1}{A} - \frac{1}{P}. \tag{41}$$

*This formula also allows for us to evaluate the perihelion shift of the planet.*

**Remark 2.** *The solution to Equation (27) is also expressed in terms of Weierstrass elliptic function* $\wp$ *as follows:*

$$\wp(\phi) = \frac{1}{P} - \frac{(A-P)([3P-4\gamma]A - 2\gamma P)}{A^2 P(P-2\gamma)\left(1 + \frac{12P}{P-2\gamma}\wp(\phi; g_2, g_3)\right)}, \tag{42}$$

*where*

$$g_2 = \frac{4\gamma^2\left(A^2 + AP + P^2\right) + 3A^2 P^2 - 6A\gamma P(A+P)}{36A^2 P^2}.$$

$$g_3 = \frac{(A-2\gamma)(2\gamma-P)(\gamma(A+P) - AP)}{216A^2 P^2}. \tag{43}$$

*The solution is periodic with period*

$$T = 2 \int_r^\infty \frac{dx}{\sqrt{4x^3 - g_2 x - g_3}}, \tag{44}$$

*where r is the greatest real root to the cubic $4x^3 - g_2 x - g_3 = 0$. This cubic has the following real roots:*

$$x_1 = \frac{\gamma}{6A} - \frac{1}{12}, x_2 = \frac{\gamma}{6P} - \frac{1}{12}, x_3 = \frac{1}{6} - \frac{\gamma(A+P)}{6AP}. \tag{45}$$

*The greatest real root is $r = x_3$; so,*

$$T = 2 \int_{x_3}^\infty \frac{dx}{\sqrt{4x^3 - g_2 x - g_3}} = \frac{2\sqrt{6AP} F\left( \sin^{-1}\left( \sqrt{2}\sqrt{-\frac{(A-P)\gamma}{3AP - 2\gamma P - 4A\gamma}} \right), \frac{2(2A+P)\gamma - 3AP}{2(A-P)\gamma} \right)}{\sqrt{\gamma}\sqrt{P-A}}. \tag{46}$$

*For Mercury (see Table 2), we have the following value from (46):*

$$T - 2\pi = 5.018176594262513 \times 10^{-7} \tag{47}$$

**Table 2.** Planets data. From [27].

| Planet | A:Apelion (m) | P:Perihelion (m) | Siederal Period | $Q = \frac{P}{A}$ |
|---|---|---|---|---|
| Mercury | 6981,7332,000 | 46,000,870,000 | 87.969089 | 0.658875 |
| Venus | 108,939,198,000 | 107,475,372,000 | 224.69562 | 0.986563 |
| Earth | 152,100,915,000 | 147,094,882,000 | 365.256622 | 0.967087 |
| Mars | 249,226,166,000 | 206,662,107,000 | 686.99329 | 0.829215 |
| Jupiter | 816,054,481,000 | 740,505,444,000 | 4334.24677 | 0.907422 |
| Saturn | 1,506,619,721,000 | 1,348,156,111,000 | 10,765.21936 | 0.894822 |
| Uranus | 3,004,984,160,000 | 2,735,977,617,000 | 30,700.24558 | 0.91048 |
| Neptune | 4,538,617,181,000 | 4,458,057,447,000 | 60,226.53638 | 0.98225 |
| Pluto | 7,377,158,662,000 | 7,437,141,859,000 | 90,631.02406 | 0.60147 |

**Remark 3.** *Solution (29) may be trigonometrically approximated in as follows. We have the approximation*

$$cn(t, m) \approx cos_m(t) := \frac{\sqrt{1+\lambda+\mu} \cos\left( \sqrt{\frac{1+\lambda+\mu}{1-\mu}} t \right)}{\sqrt{1+\lambda \cos^2\left( \sqrt{\frac{1+\lambda+\mu}{1-\mu}} t \right) + \mu \cos^4\left( \sqrt{\frac{1+\lambda+\mu}{1-\mu}} t \right)}},$$

$$\lambda = -\frac{1}{\mu+2}\left( m + 7\mu - 2m\mu - \mu^2 + m\mu^2 \right). \tag{48}$$

$$\mu = \frac{40m^2\left( 27m^2 - 128m + 128 \right)}{2409m^4 - 29600m^3 + 111520m^2 - 163840m + 81920}.$$

*Then, the approximate trigonometric solution to I.V.P. (27) reads*

$$u_{trigo}(\phi) = \frac{3}{2\gamma} - \frac{1}{P} - \frac{1}{A} + \left( \frac{1}{A} - \frac{3}{2\gamma} + \frac{2}{P} \right) cos_m^2(\omega\phi).$$

$$m = \frac{2GM(A-P)}{2GM(2A+P) - Ac^2 P}, \ \omega = \frac{1}{2}\sqrt{\frac{P-2\gamma}{P(1+m)}}, \ \gamma = \frac{3GM}{c^2}. \tag{49}$$

*Solution (49) is periodic with period*

$$T_{trigo} = \frac{\pi}{\omega}\sqrt{\frac{1-\mu}{1+\lambda+\mu}} \tag{50}$$

*For the Mercury data (see Table 2), we have*

$$\Delta_{GRT} \approx (T_{trigo} - 2\pi) \cdot \frac{23668612128}{\pi \cdot Siederal} = 42.98145640188336 \text{ arcsec/cy} \tag{51}$$

*The trigonometric solution is highly accurate.*

## 5. Contribution of the Cosmological Constant

Let us consider I.V.P. (28). The equation may be written in the form

$$u''(\phi) - \alpha + \frac{\beta}{u(\phi)^3} + u(\phi) - \gamma u(\phi)^2 = 0, \ u(0) = 1/P \text{ and } u'(0) = 0. \tag{52}$$

$$\alpha = \frac{GM}{L^2}, \beta = \frac{c^2 \Lambda}{3L^2}, \gamma = \frac{3GM}{c^2}.$$

Next, we multiply the equation $u''(\phi) - \alpha + \frac{\beta}{u(\phi)^3} + u(\phi) - \gamma u(\phi)^2 = 0$ by $u'(\phi)$ and then we integrate it with regard to $\phi$ to obtain

$$\frac{1}{2}u'(\phi)^2 - \alpha u(\phi) - \frac{\beta}{2u(\phi)^2} - \frac{1}{3}\gamma u(\phi)^3 + \frac{u(\phi)^2}{2} = C, \tag{53}$$

$C$ is the constant of integration. Letting $\phi = 0$ and taking into account conditions $u(0) = 1/P$ and $u'(0) = 0$ gives

$$C = -\frac{\gamma}{3P^3} - \frac{\beta P^2}{2} + \frac{1}{2P^2} - \frac{\alpha}{P}. \tag{54}$$

On the other hand, let $\phi_A$ be the angle for aphelion position. Then, $u(\phi_A) = 1/A$ and $u'(\phi_A) = 0$ so that

$$C = -\frac{\gamma}{3P^3} - \frac{\beta P^2}{2} + \frac{1}{2P^2} - \frac{\alpha}{P} = -\frac{\gamma}{3A^3} - \frac{A^2 \beta}{2} + \frac{1}{2A^2} - \frac{\alpha}{A}. \tag{55}$$

From (55),

$$\alpha = \frac{GM}{L^2} = \frac{3AP(A+P)(A^2\beta P^2 + 1) - 2\gamma(A^2 + AP + P^2)}{6A^2P^2}. \tag{56}$$

The angular momentum of the planet now depends on the contribution of the cosmological constant:

$$L^2 = \frac{A^2 P^2 (6GM - Ac^2\Lambda P(A+P))}{3AP(A+P) - 2\gamma(A^2 + AP + P^2)}. \tag{57}$$

From (53)–(56),

$$\left(\frac{du}{d\phi}\right)^2 = \frac{2\gamma\left(u - \frac{1}{A}\right)\left(\frac{1}{P} - u\right)\left(-u^3 + \frac{3AP - 2A\gamma - 2P\gamma}{2AP\gamma}u^2 - \frac{3AP(A+P)\beta}{2\gamma}u - \frac{3AP\beta}{2\gamma}\right)}{3u^2} \tag{58}$$

The solution to (58) is a periodic function with period

$$T_\Lambda = 2\sqrt{\frac{3}{2\gamma}} \int_{1/A}^{1/P} \frac{u\,du}{\sqrt{\left(u - \frac{1}{A}\right)\left(\frac{1}{P} - u\right)\left(-u^3 + \frac{3AP - 2A\gamma - 2P\gamma}{2AP\gamma}u^2 - \frac{3AP(A+P)\beta}{2\gamma}u - \frac{3AP\beta}{2\gamma}\right)}}. \tag{59}$$

## 6. Estimation of the Value of the Cosmological Constant

### 6.1. First Approach

Our aim is to find a value for $\Lambda$, such that the value of $T_\Lambda$ is as close as possible to value $T$ given by (40). To this end, assuming a positive cosmological constant, we obtain a range for $\Lambda$, say $10^{-k} \le \Lambda \le 10^{-m}$. Then, we evaluate (59) for $m = 1, 2, \ldots, k$. We obtain

a list of pairs $(j, T_{10-j})$ that allows us to construct an interpolating function object. This function is denoted with $\Psi$. The next step consists of minimizing the quantity $(\Psi(x) - T)^2$ on the interval $10^{-k} \leq x \leq 10^{-m}$. Let $k = 56$ and $m = 30$, so that $10^{-56} \leq \Lambda \leq 10^{-30}$. For Mercury data our calculations produced the optimal value $\Lambda_+ = 4.42355 \times 10^{-44}$. On the other hand, assuming a negative cosmological constant value, we obtained the optimal value $\Lambda_- = -7.41311 \times 10^{-42}$; see Table 3.

**Table 3.** NASA data.

| | *A*:Apelion (m) | *P*:Perihelion (m) | Siederal Period | $\hat{\Delta}_{GTR}$ |
|---|---|---|---|---|
| Mercury | $69.817 \times 10^9$ | $46.002 \times 10^9$ | 87.968 | 42.9814956752253 |
| Venus | $107.476 \times 10^9$ | $108.939 \times 10^9$ | 224.701 | 8.62485450672991 |
| Earth | $147.092 \times 10^9$ | $152.099 \times 10^9$ | 365.242 | 3.838987490384246 |
| Mars | $249.229 \times 10^9$ | $206.617 \times 10^9$ | 686.980 | 1.351057133139973 |
| Jupiter | $816.618 \times 10^9$ | $740.522 \times 10^9$ | 4332.589 | 0.062314275340345 |
| Saturn | $1514.504 \times 10^9$ | $1352.555 \times 10^9$ | 10,759.22 | 0.012639384845858 |

The confidence interval for Table 4 is:

$$
\begin{array}{ccc}
\Lambda & \Lambda_+ & \Lambda_- \\
\Delta\Lambda & \{-1.4 \times 10^{-37}, 2.3 \times 10^{-37}\} & \{-1.4 \times 10^{-37}, 2.3 \times 10^{-37}\}
\end{array}
$$

**Table 4.** Estimation of the cosmological constant using the first approach.

| Planet | $\Lambda_+$ | $\Lambda_-$ |
|---|---|---|
| Mercury | $6.56145 \times 10^{-38}$ | $-7.94328 \times 10^{-45}$ |
| Venus | $3.03249 \times 10^{-37}$ | $-10^{-46}$ |
| Earth | $4.22357 \times 10^{-38}$ | $-1.2 \times 10^{-46}$ |
| Mars | $3.8824 \times 10^{-47}$ | $-3.98 \times 10^{-39}$ |
| Jupiter | $2.02148 \times 10^{-40}$ | $-1.2 \times 10^{-48}$ |
| Saturn | $3.11889 \times 10^{-41}$ | $-1.2 \times 10^{-49}$ |
| Uranus | $3.98107 \times 10^{-42}$ | $-10^{-50}$ |
| Neptune | $1.58489 \times 10^{-42}$ | $-1.26 \times 10^{-51}$ |
| Pluto | $5.12861 \times 10^{-44}$ | $-2.5 \times 10^{-51}$ |

### 6.2. Second Approach

Integral (59) is hard to evaluate, so we give an approximate analytical expression. Assume a positive cosmological constant $\Lambda = \Lambda_+$. We have

$$
\begin{aligned}
&-u^3 + \frac{3AP - 2A\gamma - 2P\gamma}{2AP\gamma}u^2 - \frac{3AP(A+P)\beta}{2\gamma}u - \frac{3AP\beta}{2\gamma} < \\
&-u^3 + \frac{3AP - 2A\gamma - 2P\gamma}{2AP\gamma}u^2 - \frac{3AP(A+P)\beta}{2\gamma}u < \\
&-u^3 + \frac{3AP - 2A\gamma - 2P\gamma}{2AP\gamma}u^2 - \frac{3AP(A+P)\beta}{2\gamma}u^2 = \\
&u^2(\delta_1 - u), \text{ where} \\
&\delta_1 = \frac{3AP - 2A\gamma - 2P\gamma}{2AP\gamma} - \frac{3AP(A+P)\beta}{2\gamma}. \text{ Then}
\end{aligned}
\tag{60}
$$

$$
\frac{u}{\sqrt{-u^3 + \frac{3AP - 2A\gamma - 2P\gamma}{2AP\gamma}u^2 - \frac{3AP(A+P)\beta}{2\gamma}u - \frac{3AP\beta}{2\gamma}}} > \frac{1}{\sqrt{\delta_1 - u}}.
$$

On the other hand, since $u \geq 1/A$, $u^{-1} \leq A$, $-u^{-1} \geq -A$ and $-u^{-2} \geq -A^2$, and then

$$
\begin{aligned}
-u^3 &+ \frac{3AP-2A\gamma-2P\gamma}{2AP\gamma}u^2 - \frac{3AP(A+P)\beta}{2\gamma}u - \frac{3AP\beta}{2\gamma} = \\
&u^2\left(-u + \frac{3AP-2A\gamma-2P\gamma}{2AP\gamma} - \frac{3AP(A+P)\beta}{2\gamma}u^{-1} - \frac{3AP\beta}{2\gamma}u^{-2}\right) \geq \\
&u^2\left(-u + \frac{3AP-2A\gamma-2P\gamma}{2AP\gamma} - \frac{3AP(A+P)\beta}{2\gamma}A - \frac{3AP\beta}{2\gamma}A^2\right) = \\
&u^2(\delta_2 - u), \text{ where} \\
\delta_2 &= \frac{3AP-2A\gamma-2P\gamma}{2AP\gamma} - \frac{3AP(A+P)\beta}{2\gamma}A - \frac{3AP\beta}{2\gamma}A^2. \text{ Then}
\end{aligned}
\tag{61}
$$

$$
\frac{u}{\sqrt{-u^3 + \frac{3AP-2A\gamma-2P\gamma}{2AP\gamma}u^2 - \frac{3AP(A+P)\beta}{2\gamma}u - \frac{3AP\beta}{2\gamma}}} \leq \frac{1}{\sqrt{\delta_2 - u}}.
$$

From (60) and (61), we obtain

$$
\frac{1}{\sqrt{\delta_1 - u}} \leq \frac{u}{\sqrt{-u^3 + \frac{3AP-2A\gamma-2P\gamma}{2AP\gamma}u^2 - \frac{3AP(A+P)\beta}{2\gamma}u - \frac{3AP\beta}{2\gamma}}} < \frac{1}{\sqrt{\delta_2 - u}},
\tag{62}
$$

Then, taking into account (59), we have estimates

$$
F(\delta_1) \leq T_{\Lambda_+} < F(\delta_2),
\tag{63}
$$

where

$$
F(\delta) := 2\sqrt{\frac{3}{2\gamma}} \int_{1/A}^{1/P} \frac{du}{\sqrt{\left(u - \frac{1}{A}\right)\left(\frac{1}{P} - u\right)(\delta - u)}} = \frac{2\sqrt{6A}K\left(\frac{A-P}{P(A\delta - 1)}\right)}{\sqrt{\gamma}\sqrt{A\delta - 1}}.
\tag{64}
$$

Then,

$$
T_{\Lambda_+} \approx \frac{1}{2}(F(\delta_1) + F(\delta_2)).
\tag{65}
$$

Equating (40) and (65), we obtain the required equation to determine $\Lambda_+$.
On the other hand, suppose that $\Lambda = \Lambda_-$ is negative. Then, since $u > u^2$,

$$
\begin{aligned}
-u^3 &+ \frac{3AP-2A\gamma-2P\gamma}{2AP\gamma}u^2 - \frac{3AP(A+P)\beta}{2\gamma}u - \frac{3AP\beta}{2\gamma} > \\
-u^3 &+ \frac{3AP-2A\gamma-2P\gamma}{2AP\gamma}u^2 - \frac{3AP(A+P)\beta}{2\gamma}u > \\
-u^3 &+ \frac{3AP-2A\gamma-2P\gamma}{2AP\gamma}u^2 - \frac{3AP(A+P)\beta}{2\gamma}u^2 = \\
&u^2\left(\frac{3AP-2A\gamma-2P\gamma}{2AP\gamma} - \frac{3AP(A+P)\beta}{2\gamma} - u\right) = \\
&u^2(\delta_4 - u), \delta_4 = \frac{3AP-2A\gamma-2P\gamma}{2AP\gamma} - \frac{3AP(A+P)\beta}{2\gamma}.
\end{aligned}
\tag{66}
$$

so that

$$
\frac{u}{\sqrt{-u^3 + \frac{3AP-2A\gamma-2P\gamma}{2AP\gamma}u^2 - \frac{3AP(A+P)\beta}{2\gamma}u - \frac{3AP\beta}{2\gamma}}} < \frac{1}{\sqrt{\delta_4 - u}}
\tag{67}
$$

Now, since $u^{-1} \leq A$,

$$
\begin{aligned}
-u^3 &+ \frac{3AP-2A\gamma-2P\gamma}{2AP\gamma}u^2 - \frac{3AP(A+P)\beta}{2\gamma}u - \frac{3AP\beta}{2\gamma} = \\
&u^2\left(-u + \frac{3AP-2A\gamma-2P\gamma}{2AP\gamma} - \frac{3AP(A+P)\beta}{2\gamma}u^{-1} - \frac{3AP\beta}{2\gamma}u^{-2}\right) \leq \\
&u^2\left(-u + \frac{3AP-2A\gamma-2P\gamma}{2AP\gamma} - \frac{3AP(A+P)\beta}{2\gamma}A - \frac{3AP\beta}{2\gamma}A^2\right) = \\
&u^2(\delta_3 - u), \text{ where} \\
\delta_3 &= \frac{3AP-2A\gamma-2P\gamma}{2AP\gamma} - \frac{3AP(A+P)\beta}{2\gamma}A - \frac{3AP\beta}{2\gamma}A^2.
\end{aligned}
\tag{68}
$$

We obtained the following estimates:

$$\frac{1}{\sqrt{\delta_3 - u}} \leq \frac{u}{\sqrt{-u^3 + \frac{3AP - 2A\gamma - 2P\gamma}{2AP\gamma}u^2 - \frac{3AP(A+P)\beta}{2\gamma}u - \frac{3AP\beta}{2\gamma}}} \tag{69}$$

Using (68) and (69), and taking into account (59)–(64) we have:

$$F(\delta_3) \leq T_{\Lambda_-} < F(\delta_4). \tag{70}$$

The equation for $\Lambda_-$ is $\frac{1}{2}(F(\delta_3) + F(\delta_4)) = T$, where $T$ is found from (40).

## 7. Approximate Analytical Solution to the Motion Equation with a Nonzero Cosmological Constant

### 7.1. First Solution Method

Since $\Lambda$ is small,

$$u^3 + \frac{3AP - 2A\gamma - 2P\gamma}{2AP\gamma}u^2 - \frac{3AP(A+P)\beta}{2\gamma}u - \frac{3AP\beta}{2\gamma} \approx u^3 + \frac{3AP - 2A\gamma - 2P\gamma}{2AP\gamma}u^2 - \frac{3AP(A+P)\beta}{2\gamma}u^2. \tag{71}$$

Then , Hard Ode (58) may be approximated by the ode

$$\left(\frac{du}{d\phi}\right)^2 = \frac{2\gamma}{3}\left(u - \frac{1}{A}\right)\left(\frac{1}{P} - u\right)(\delta - u), \text{ where}$$

$$\delta = \frac{3AP - 2A\gamma - 2P\gamma}{2AP\gamma} - \frac{3AP(A+P)\beta}{2\gamma}, \beta = \frac{c^2\Lambda}{3L^2}. \tag{72}$$

$$L^2 = \frac{A^2P^2\left(6GM - Ac^2\Lambda P(A+P)\right)}{3AP(A+P) - 2\gamma(A^2 + AP + P^2)}.$$

The exact solution to Ode (72) given initial conditions $u(0) = 1/P$ and $u'(0) = 0$ is given by

$$u(\phi) = \delta + \left(\frac{1}{P} - \delta\right)\text{nd}\left(\sqrt{\frac{\gamma(A\delta - 1)}{6A}}\phi, \frac{A - P}{P(A\delta - 1)}\right)^2 \tag{73}$$

Let us compare this solution with the numerical solution to equivalent Ode (28) for Mercury data assuming the value of $\Lambda = 10^{-46}$, see Figure 2.

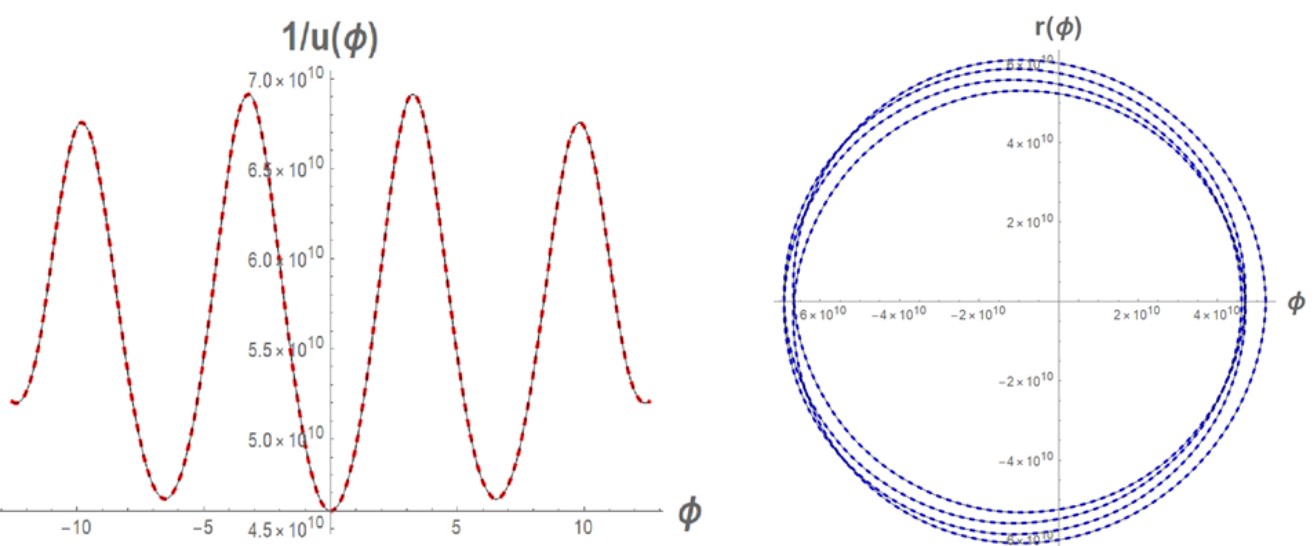

**Figure 2.** Comparison between semianalytical (73) and numerical solutions to the ode.

Our semianalytical solution was highly accurate and involved the cosmological constant. This also offers another way to estimate the value of the cosmological constant. As a matter of fact, the period of the semianalytical solution reads

$$T_\Lambda = 2 \frac{K\left(\frac{A-P}{P(A\delta-1)}\right)}{\sqrt{\frac{\gamma(A\delta-1)}{6A}}} \tag{74}$$

Then, the equation for $\Lambda$ is

$$T_\Lambda = 2\sqrt{\frac{6A}{\gamma(A\delta-1)}}K\left(\frac{A-P}{P(A\delta-1)}\right) = T := \frac{4K\left(\frac{2GM(A-P)}{Ac^2P-2GM(A+2P)}\right)}{\sqrt{1-\frac{2GM(A+2P)}{Ac^2P}}} \tag{75}$$

Since the value of $m = \frac{A-P}{P(A\delta-1)}$ is small, we may use the following approximation:

$$2K(m) \mathrel{\overset{\circ}{=}} \frac{1-\frac{5m}{16}}{1-\frac{9m}{16}}\pi. \tag{76}$$

Then,

$$T_\Lambda = \frac{\sqrt{6}\pi\sqrt{A}(A(16\delta P-5)-11P)}{\sqrt{\gamma}\sqrt{A\delta-1}(A(16\delta P-9)-7P)} = T, \tag{77}$$

from where

$$\begin{array}{l} 256A^3P^2T^2\gamma\delta^3 - 96A^2P\left(16AP\pi^2 + 3AT^2\gamma + 5PT^2\gamma\right)\delta \\ +3A\left(320A^2P\pi^2 + 704AP^2\pi^2 + 27A^2T^2\gamma + 138APT^2\gamma + 91P^2T^2\gamma\right)\delta^2 \\ -150A^3\pi^2 - 660A^2P\pi^2 - 726AP^2\pi^2 - 81A^2T^2\gamma - 126APT^2\gamma - 49P^2T^2\gamma = 0. \end{array} \tag{78}$$

Solving this cubic, we obtain the value for $\delta$. Then, from (72),

$$\Lambda = \frac{6\mu Q^2(2\gamma\delta P - 3P + 2\gamma(Q+1))}{c^2P^2(Q+1)(2\gamma + 2\gamma\delta P^2 + 2\gamma(Q+1)(P+Q) - 3P(P+Q+1))}, Q := \frac{P}{A}. \tag{79}$$

We may go further using other approximates for $K(m)$. For example,

$$K(m) \mathrel{\overset{\circ}{=}} \frac{\pi\left(409m^2 - 3984m + 4864\right)}{1025m^2 - 5200m + 4864} \tag{80}$$

This approximation gives us the following quintic for $\delta$:

$$C_0\delta^5 + C_1\delta^4 + C_2\delta^3 + C_3\delta^2 + C_4\delta + C_5 = 0, \tag{81}$$

where

$$C_0 = 23658496A^5\gamma P^4T^2. \tag{82}$$

$$C_1 = -155648A^4P^3\left(912\pi^2 AP + 325A\gamma T^2 + 435\gamma PT^2\right). \tag{83}$$

$$C_2 = 256A^3P^2\left(908352\pi^2 A^2P + 144575A^2\gamma T^2 + 1309632\pi^2 AP^2 + 501250A\gamma PT^2 + 278335\gamma P^2T^2\right). \tag{83}$$

$$C_3 = -32A^2P\left(\begin{array}{l}3722064\pi^2 A^3P + 333125A^3\gamma T^2 + 14356320\pi^2 A^2P^2 + 2470425A^2\gamma PT^2 + \\ 8537424\pi^2 AP^3 + 3544575A\gamma P^2T^2 + 1045155\gamma P^3T^2\end{array}\right). \tag{84}$$

$$C_4 = A\left(\begin{array}{l}19553472\pi^2 A^4P + 1050625A^4\gamma T^2 + 179551680\pi^2 A^3P^2 + 17117500A^3\gamma PT^2 + 279850560\pi^2 A^2P^3 + \\ 53377350A^2\gamma P^2T^2 + 88848192\pi^2 AP^4 + 40032700A\gamma P^3T^2 + 6714305\gamma P^4T^2\end{array}\right). \tag{85}$$

$$C_5 = -1003686\pi^2 A^5 - 15538728\pi^2 A^4 P - 1050625 A^4 \gamma T^2 - 66467748\pi^2 A^3 P^2 - 6457500 A^3 \gamma P T^2 - \tag{86}$$
$$48971688\pi^2 A^2 P^3 - 11334950 A^2 \gamma P^2 T^2 - 9969126\pi^2 A P^4 - 4340700 A \gamma P^3 T^2 - 474721 \gamma P^4 T^2.$$

Solving the quintic, we obtain one or more estimates for $\Lambda$. Let us evaluate the $\Lambda$ using Formula (79) for Mercury. The cubic resolvent reads

$$-3.22413405750441056780 8405 \times 10^{61} \delta^3 + 1.09172401507092103117671711214 \times 10^{58} \delta^2 -$$
$$3.63336093243590128854 22051569104 \times 10^{47} \delta + 3.02304248389437670168757263812 36 \times 10^{36} = 0. \tag{87}$$

Its roots are:

$$\delta_1 = 1.66403214678195905033219994 \times 10^{-11}.$$
$$\delta_2 = 1.66406282054722562545891615 \times 10^{-11}. \tag{88}$$
$$\delta_3 = 0.000338609961092532598 9588409.$$

The values of $\Lambda$ from (79) are:

$$\Lambda_1 = 2.38185525520683097701 72818 \times 10^{-29}. \tag{89}$$

$$\Lambda_2 = 2.38185525520683097701 72040 \times 10^{-29} \tag{90}$$

$$\Lambda_3 = 2.574519419707803 \times 10^{-43} \tag{91}$$

Using the quintic, we obtain the negative value

$$\Lambda = -1.024109331810223183 3808 \times 10^{-47} \tag{92}$$

The algebraic approach may give either positive or negative estimates for the cosmological constant. Letting $\Lambda = 2.5 \times 10^{-43}$, we obtain the following polar plot (Figure 3) for Mercury's trajectory using the numerical solution to Ode (28).

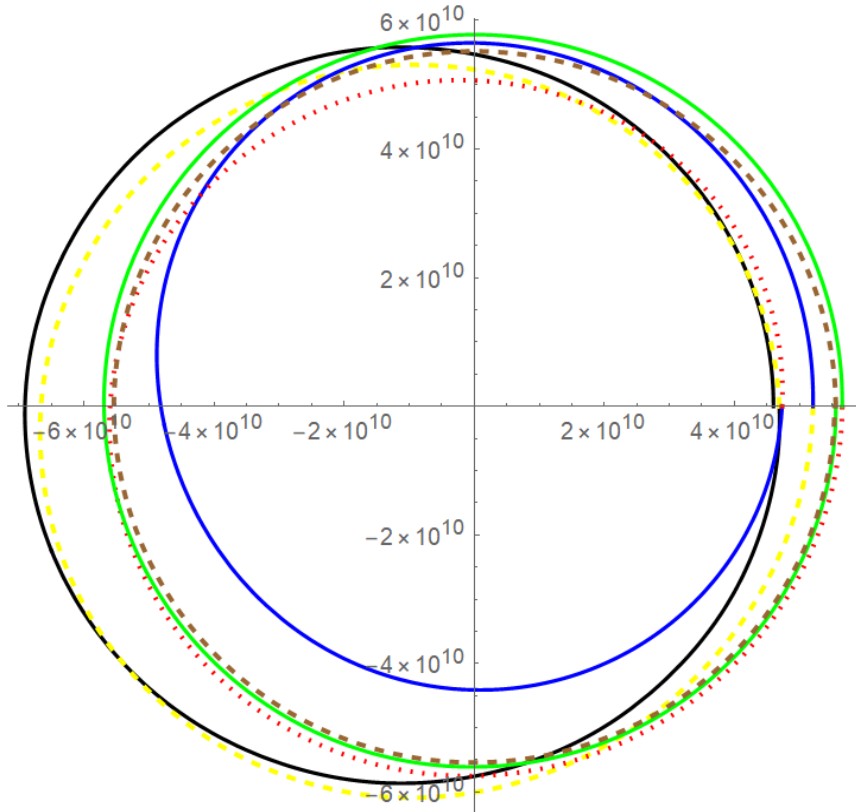

**Figure 3.** Mercury's trajectory using Ode (28) with $\Lambda = \Lambda_+ = 2.57 \times 10^{-43}$.

After many calculations to obtain $\Lambda$ estimates, we found the following interesting fact. Number $\delta$ in Equation (72) was very close to $\delta = 1/2953$ for all planets in the solar system. The mean of these values was

$$\delta = 0.0003386099880148952121999968489565.$$

This implies that

$$\frac{\sqrt{6}K\left(\frac{A-P}{P(A\delta-1)}\right)}{\sqrt{\gamma\left(\delta - \frac{1}{A}\right)}} = \frac{2K\left(\frac{2(A-P)\mu}{2(2A+P)\mu-Ac^2P}\right)}{\sqrt{1 - \frac{2\mu(2A+P)}{Ac^2P}}}, \mu = GM. \tag{93}$$

That is, the planets are related to Curve (93) in the $A - P$ plane. This allows for us to *predict* the value of the aphelion given the the perihelion and vice versa. Indeed, if we are given the aphelion, we simply solve transcendental Equation (93) for the perihelion. On the other hand, the fact that $\delta$ is almost a constant tells us that $\Lambda$ defined by (79) must indeed be a *constant* !. This justifies the name 'cosmological constant'; see Figure 4.

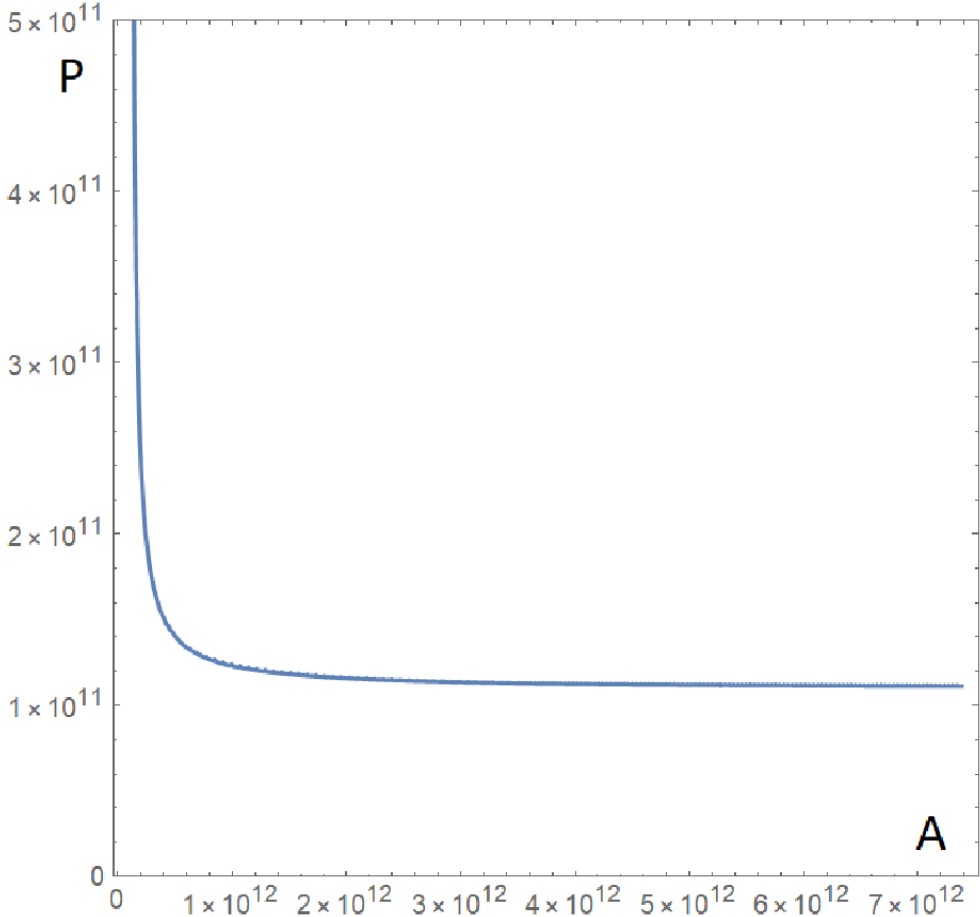

**Figure 4.** Curve (93).

### 7.2. Second Solution Method

Let

$$F(u) = -u^3 + \frac{3AP - 2A\gamma - 2P\gamma}{2AP\gamma}u^2 - \frac{3AP(A+P)\beta}{2\gamma}u - \frac{3AP\beta}{2\gamma}, \quad \beta = \frac{c^2\Lambda}{3L^2}. \tag{94}$$

Let us introduce the following notations:

$$d_0 = -\frac{3AP\beta}{2\gamma}, \quad d_1 = -\frac{3AP(A+P)\beta}{2\gamma}, \quad d_2 = \frac{3AP - 2A\gamma - 2P\gamma}{2AP\gamma} \tag{95}$$

For a positive $\Lambda$, the cubic $d_0 + d_1u + d_2u^2 - u^3 = 0$ has two small real roots of opposite signs. For a negative $\Lambda$, we have only one positive real root, and the two other roots are imaginary and small in magnitude. This allows for us to perform the approximation

$$F(u) \approx u^2 \left( \frac{d_2^4 + 3d_1d_2^2 + 2d_0d_2 + d_1^2}{d_2^3 + 2d_1d_2 + d_0} - u \right). \tag{96}$$

Then, hard Ode (58) is replaced with easy ode

$$\left( \frac{du}{d\phi} \right)^2 = \frac{2\gamma}{3} \left( u - \frac{1}{A} \right) \left( \frac{1}{P} - u \right) \left( \frac{d_2^4 + 3d_1d_2^2 + 2d_0d_2 + d_1^2}{d_2^3 + 2d_1d_2 + d_0} - u \right) \tag{97}$$

so that, in view of (64),

$$T_\Lambda \approx 2\sqrt{\frac{3}{2\gamma}} \int_{1/A}^{1/P} \frac{du}{\sqrt{\left( u - \frac{1}{A} \right) \left( \frac{1}{P} - u \right) (\delta - u)}}, \tag{98}$$

$$\delta = \frac{d_2^4 + 3d_1d_2^2 + 2d_0d_2 + d_1^2}{d_2^3 + 2d_1d_2 + d_0}$$

The required equation to estimate the $\Lambda$ is then

$$2\sqrt{\frac{6A}{\gamma(A\delta - 1)}} K\left( \frac{A - P}{P(A\delta - 1)} \right) = \frac{4K\left( \frac{2GM(A-P)}{Ac^2P - 2GM(A+2P)} \right)}{\sqrt{1 - \frac{2GM(A+2P)}{Ac^2P}}}. \tag{99}$$

For Mercury data, the estimated value using (99) is $\Lambda_+ = 6.51 \times 10^{-52}$. For Pluto, $\Lambda_- = -2.83 \times 10^{-52}$. For data [25] $P = 46005766800$, $A = 69818681600$, the estimated value using Formula (99) is $\Lambda = -3.52 \times 10^{-51}$.

### 7.3. Third Approach

We may proceed quite differently in order to estimate the $\Lambda$ value. Let $u_{\text{app}}(\phi)$ be some approximate solution to I.V.P. (27), for example, $u = u_{\text{trigo}}$. Then, replacing $u$ with $u_{\text{app}}$ in (58) gives

$$\Lambda(\phi) = f(\phi) := \frac{L_0^2 u_{\text{app}}(\phi)^2 \left( \begin{array}{l} 3AP\left( (Au_{\text{app}}(\phi) - 1)(Pu_{\text{app}}(\phi) - 1) - APu_{\text{app}}'(\phi)^2 \right) - \\ 2\gamma (Au_{\text{app}}(\phi) - 1)(Pu_{\text{app}}(\phi) - 1)(APu_{\text{app}}(\phi) + A + P) \end{array} \right)}{A^2 c^2 P^2 (Au_{\text{app}}(\phi) - 1)(Pu_{\text{app}}(\phi) - 1)((A+P)u_{\text{app}}(\phi) + 1)}. \tag{100}$$

Then, we take

$$\Lambda_{\text{integral}} = \Lambda(A, P) = \frac{1}{T_0} \int_{-T_0/2}^{T_0/2} f'(\phi) d\phi. \tag{101}$$

See Table 5 for the different $\Lambda$ estimates.

**Table 5.** $\delta$ value in Equation (72) .

| | |
|---|---|
| Mercury | $\delta = 0.00033860996109253251387660732518 7$ |
| Venus | $\delta = 0.00033860997867045345777539666620 7$ |
| Earth | $\delta = 0.00033860998378142733063189684372 7$ |
| Mars | $\delta = 0.00033860998830310685666147851868 9$ |
| Jupiter | $\delta = 0.00033860999457850535178035644712 2$ |
| Saturn | $\delta = 0.00033860999574885156104819827760 4$ |
| Uranus | $\delta = 0.00033860999645606238090192463552 3$ |
| Neptune | $\delta = 0.00033860999670969841451872994753 1$ |
| Pluto | $\delta = 0.00033860999679341904260512774449 4$ |

Predicted value for $\Lambda_+$:

$$\text{Arithmetic Mean: } \Lambda_+ = 2.87 \times 10^{-44}.$$
$$\text{Geometric Mean: } \Lambda_+ = 1.25 \times 10^{-52} \tag{102}$$

Predicted value for $\Lambda_-$:

$$\text{Arithmetic mean: } \Lambda_- = -7.25 \times 10^{-44}.$$
$$\text{Geometric mean: } \Lambda_- = -1.01 \times 10^{-52} \tag{103}$$

Confidence interval for Table 6:

| $\Lambda$ | $\Lambda_+$ | $\Lambda_-$ |
|---|---|---|
| $\Delta\Lambda$ | $\{-1.3 \times 10^{-43}, 1.9 \times 10^{-43}\}$ | $\{-4.8 \times 10^{-43}, 3.3 \times 10^{-43}\}$ |

**Table 6.** Estimation of the cosmological constant using the second and third approaches.

| Planet | $\Lambda_+$ | $\Lambda_-$ | $\Lambda_{\text{integral}}$ |
|---|---|---|---|
| Mercury | $2.58 \times 10^{-43}$ | $-1.02 \times 10^{-47}$ | $4.18351 \times 10^{-37}$ |
| Venus | $6.34 \times 10^{-49}$ | $-6.51 \times 10^{-43}$ | $5.54028 \times 10^{-38}$ |
| Earth | $1.05 \times 10^{-49}$ | $-4.26 \times 10^{-45}$ | $1.45838 \times 10^{-38}$ |
| Mars | $4.0 \times 10^{-51}$ | $-5.35 \times 10^{-45}$ | $-2.22584 \times 10^{-39}$ |
| Jupiter | $3.33 \times 10^{-53}$ | $-2.46 \times 10^{-55}$ | $-2.62516 \times 10^{-41}$ |
| Saturn | $5.08 \times 10^{-55}$ | $-3.45 \times 10^{-60}$ | $-2.53073 \times 10^{-42}$ |
| Uranus | $2.8 \times 10^{-55}$ | $-5.33 \times 10^{-59}$ | $1.42783 \times 10^{-43}$ |
| Neptune | $4.13 \times 10^{-56}$ | $-2.00 \times 10^{-59}$ | $2.13098 \times 10^{-44}$ |
| Pluto | $1.38 \times 10^{-56}$ | $-8.303 \times 10^{-60}$ | $3.78788 \times 10^{-45}$ |

## 8. Analysis and Discussion

We obtained the estimated values for the cosmological constant, and we compared our results with that obtained in [25]. The nonlinear differential equation in [25] taking into account the cosmological constant reads

$$\left(\frac{du}{d\phi}\right)^2 = \frac{2GM}{c^2 u(\phi)^2}\left(u(\phi) - \frac{1}{A}\right)\left(\frac{1}{P} - u(\phi)\right)\left(d_0 + d_1 u(\phi) + d_2 u(\phi)^2 - u^3(\phi)\right), \tag{104}$$

where

$$d_0 = \frac{c^2\left(A^2\left(c^2 P - 2GM\right) + AP\left(c^2 P - 2GM\right) - 2GMP^2\right)}{2AGMP(Ac^2\Lambda P(A+P) + 6GM)}\Lambda.$$

$$d_1 = \frac{c^2(A+P)\left(A^2\left(c^2 P - 2GM\right) + AP\left(c^2 P - 2GM\right) - 2GMP^2\right)}{2AGMP(Ac^2\Lambda P(A+P) + 6GM)}\Lambda. \tag{105}$$

$$d_2 = \frac{c^2}{2GM} - \frac{1}{A} - \frac{1}{P}.$$

In the case when $\Lambda = 0$ the exact solution is

$$u(\phi) = \frac{c^2}{6GM} + \frac{2c^2}{GM} \wp(t + t_0; g_2, g_2),$$

where

$$g_2 = g_2 = \frac{A^2c^4P^2 - 6A^2c^2GMP + 12A^2G^2M^2 - 6Ac^2GMP^2 + 12AG^2M^2P + 12G^2M^2P^2}{12A^2c^4P^2}.$$

$$g_3 = g_3 = \frac{\left(Ac^2 - 6GM\right)\left(c^2P - 6GM\right)\left(Ac^2P - 3AGM - 3GMP\right)}{216A^2c^6P^2}.$$

$$t_0 = \wp^{-1}\left(\frac{GM}{2c^2}\left(\frac{1}{P} - \frac{c^2}{6GM}\right); g_2, g_2\right)$$

The solution is periodic, and its period is given by

$$T = \frac{4K\left(\frac{2GM(A-P)}{Ac^2P - 2GM(A+2P)}\right)}{\sqrt{1 - \frac{2GM(A+2P)}{Ac^2P}}}.$$

In [25] authors use the following data:

$$P = 45997620600, \quad A = 69820729300. \tag{106}$$

The perihelion shift for these data is 42.9825 arc-sec/century. The value in [25] was =42.9817 arc-sec/century. On the other hand, for the data in [25],

$$P = 46001260500, \quad A = 69817089400. \tag{107}$$

The perihelion shift for these data is 42.9814 arc-sec/century. The value in [25] is 42.9805 arc-sec/century. The authors in [25] used the data

$$P = 46010448900, \quad A = 69807901000. \tag{108}$$

For these data, the perihelion shift is 42.9784 arc-sec/century. The value in [25] is =42.9776 arc-sec/century.

Using Mercury data in Table 2, we obtain a perihelion shift of 42.9815 arc-sec/century.

In the case when $\Lambda \neq 0$, cubic $d_0 + d_1u + d_2u^2 - u^3 = 0$ has two roots very close to zero. Then, we may use the approximation

$$d_2u^2 + d_1u + d_0 - u^3 \approx \left(\frac{d_0\left(d_2^2 + d_1\right)}{d_2^3 + 2d_1d_2 + d_0} + \frac{\left(d_1^2 + d_2^2d_1 + d_0d_2\right)}{d_2^3 + 2d_1d_2 + d_0}u + u^2\right)\left(\frac{d_2^4 + 3d_1d_2^2 + 2d_0d_2 + d_1^2}{d_2^3 + 2d_1d_2 + d_0} - u\right) \tag{109}$$

For example, let $\Lambda = -10^{-56}$, $P = 46005766800$, $A = 69818681600$ as in [25]. We have:

$$\begin{aligned} d_0 + d_1u + d_2u^2 - u^3 = \quad &(u - 3.6156749990681577 \times 10^{-25}) \\ &(u + 3.6156749990680067 \times 10^{-25}) \\ &(0.0003386099606939169 - u). \end{aligned} \tag{110}$$

On the other hand, the right-hand side of (109) is written as

$$(u - 3.6156749990681577 \times 10^{-25})(u + 3.6156749990680067 \times 10^{-25})(0.00033860996069391686 - u) \tag{111}$$

Using these facts and taking into account that

$$\frac{1}{u^2}\left[\frac{d_0\left(d_2^2+d_1\right)}{d_2^3+2d_1d_2+d_0}+\frac{\left(d_1^2+d_2^2d_1+d_0d_2\right)}{d_2^3+2d_1d_2+d_0}u+u^2\right]$$
$$\approx\frac{1}{A^2}\left[\frac{d_0\left(d_2^2+d_1\right)}{d_2^3+2d_1d_2+d_0}+\frac{\left(d_1^2+d_2^2d_1+d_0d_2\right)}{d_2^3+2d_1d_2+d_0}\cdot\frac{1}{A}+\frac{1}{A^2}\right]:=N \tag{112}$$

hard Ode (104) may be replaced with easy ode

$$\left(\frac{du}{d\phi}\right)^2=D\left(u(\phi)-\frac{1}{A}\right)\left(\frac{1}{P}-u(\phi)\right)\left(\frac{d_2^4+3d_1d_2^2+2d_0d_2+d_1^2}{d_2^3+2d_1d_2+d_0}-u(\phi)\right). \tag{113}$$

The exact solution to Ode (113) is

$$u(\phi)=\frac{1}{P}+\frac{3(P-A)(\delta P-1)}{P(A\delta P-2A+P)\left(1+\frac{12AP}{D(A\delta P-2A+P)}\wp(t;g_2,g_3)\right)}, \tag{114}$$

where

$$D=\frac{2GMN}{c^2},$$
$$\delta=\frac{d_2^4+3d_1d_2^2+2d_0d_2+d_1^2}{d_2^3+2d_1d_2+d_0}. \tag{115}$$

$$g_2=\frac{D^2\left(A^2\delta^2P^2-A^2\delta P+A^2-A\delta P^2-AP+P^2\right)}{12A^2P^2}.$$

$$g_3=\frac{D^3(A\delta P+A-2P)(A\delta P-2A+P)(2A\delta P-A-P)}{432A^3P^3}.$$

Solution (113) is periodic, and its period equals

$$T_\Lambda=2\int_r^\infty\frac{1}{\sqrt{4x^3-g_2x-g_3}}=2\sqrt{\frac{P}{D(\delta P-1)}}K\left(-\frac{A-P}{A(P\delta-1)}\right), \tag{116}$$

where $r$ is the greatest real root to cubic $4x^3-g_2x-g_3=0$. Observe that

$$4x^3-g_2x-g_3=\left(x+\frac{D(A(1+P\delta)-2P)}{12AP}\right)\left(x+\frac{D(P(1+A\delta)-2A)}{12AP}\right)\left(x-\frac{D(2AP\delta-A-P)}{12AP}\right). \tag{117}$$

The required equation to determine $\Lambda$ is

$$2\sqrt{\frac{P}{D(\delta P-1)}}K\left(\frac{A-P}{A(1-P\delta)}\right)=T:=\frac{4K\left(\frac{2GM(A-P)}{2GM(2A+P)-Ac^2P}\right)}{\sqrt{1-\frac{2GM(2A+P)}{Ac^2P}}} \tag{118}$$

The authors in [25] gave the predicted values $\Lambda_-=-10^{-55}$ and $\Lambda_+=10^{56}$ for the data

$$P=46005766800,\ A=69818681600. \tag{119}$$

On the other hand, the predicted values for $\Lambda$ solving (118) for data (119) are

$$\Lambda_+=3.0849194019988974\times10^{-45}\text{ and }\Lambda_-=-1.8381582529562642\times10^{-45} \tag{120}$$

Now, using the method in previous sections, we obtained the following estimates for Data (119) in [25]

$$\Lambda_+=1.024\times10^{-47}\text{ and }\Lambda_-=-2.58\times10^{-43} \tag{121}$$

Lastly, for the other planets, the estimates for $\Lambda$ by solving (118) are summarized in Table 7:

**Table 7.** Theoretical values for $\Lambda$.

| Planet | $\Lambda_+$ | $\Lambda_-$ |
|---|---|---|
| Mercury | $3.09 \times 10^{-45}$ | $-1.83 \times 10^{-45}$ |
| Venus | $2.64 \times 10^{-48}$ | $-2.56 \times 10^{-47}$ |
| Earth | $1.46 \times 10^{-47}$ | $-2.12 \times 10^{-46}$ |
| Mars | $8.71 \times 10^{-48}$ | $-7.56 \times 10^{-32}$ |
| Jupiter | $9.98 \times 10^{-49}$ | $-1.885 \times 10^{-33}$ |
| Saturn | $8.01 \times 10^{-50}$ | $-2.74 \times 10^{-51}$ |
| Uranus | $2.23 \times 10^{-51}$ | $-1.99 \times 10^{-50}$ |
| Neptune | $2.18 \times 10^{-37}$ | $-9.1 \times 10^{-51}$ |
| Pluto | $3.68 \times 10^{-51}$ | $-4.66 \times 10^{-36}$ |

Confidence interval for Table 7:

| $\Lambda$ | $\Lambda_+$ | $\Lambda_-$ |
|---|---|---|
| $\Delta\Lambda$ | $\{-1.1 \times 10^{-37}, 1.6 \times 10^{-37}\}$ | $\{-5.5 \times 10^{-32}, 3.8 \times 10^{-32}\}$ |

The different previous approaches show that the cosmological constant takes different values, both positive and negative, which leads to establishing that the static cosmological model is unstable and antisymmetric. Due to the difference in the perihelion for each planet, the cosmological constant is different for each one, which implies that the trajectory of the planets must be antisymmetric, that is, elliptical.

The Schwarzschild radius for the Sun is $r_S = \frac{2GM}{c^2} = 2.7$ km. Taking into account the cosmological constant, the Schwarzschild radius is obtained by solving the following cubic equation for $r$.

$$1 - \frac{1}{3}r^2\Lambda - \frac{2GM}{c^2 r} = 0. \tag{122}$$

Solving (122), we obtain that $r_{S\Lambda} = 4.1$ km. Both radii are inside the Sun.

The region of spacetime due to the gravitational field of the Sun is obtained from

$$1 - \frac{1}{3}r^2\Lambda - \frac{2GM}{c^2 r} = 1, \tag{123}$$

from where $r_{\text{lim}} = 5.1 \times 10^{10}$ Km = 340 UA.

Between the planet Pluto and the edge of the curved space, an area of the curved spacetime region originates that allows for, in Schwarzschild's cosmological model, predicting the existence of new planets (planetoids $X$). Among those catalogued planets are: Kuiper, whose distance to the Sun is 30 AU; Quaoar, 43.4 AU; Makameke, 45.8 AU; Sedna, 88.5 AU; Eris, 98.3 AU; and V774104, 103 AU.

This limit was calculated for the cosmological constant obtained for Mercury $\Lambda = 6.56145 \times 10^{-38} \text{m}^{-2}$.

After the boundary of the curved spacetime, a region of Minkowski plane spacetime originates.

From Schwarzschild's stationary cosmological model (Figure 5), it follows that the planets are in the region of spacetime curved due to the Sun; therefore, each planet follows a stationary geodesic trajectory. All of the above are represented in the following graph.

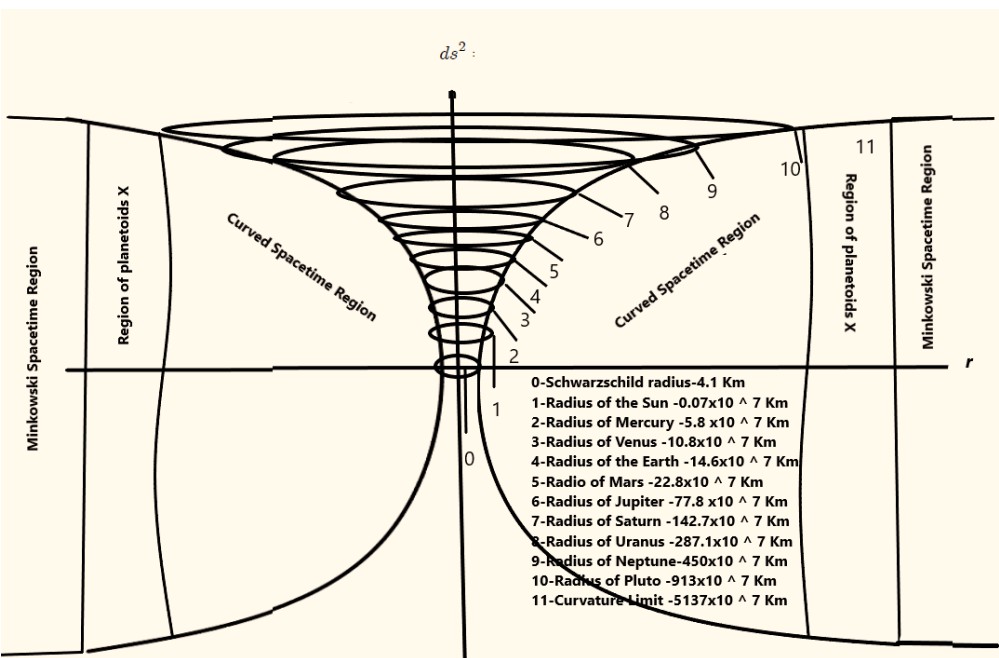

**Figure 5.** Schwarzschild's cosmological model .

The static cosmological model of Schwarzschild, the cosmological constant according to the results obtained in Tables 4 and 6, which were obtained by different approaches, shows that, in the planetary system, the cosmological constant is unstable. In order to understand this effect of instability in the static cosmological model, it is necessary to take into account the effects of the acceleration due to the Big Bang until the current static state is reached.

Lastly, the instability effect of the cosmological constant leads to the fluctuation of the limit radius of the curved spacetime, affecting the stability of the curvature region.

## 9. Conclusions

In this work, the equation of Einstein's general theory of relativity for vacuum was solved. Taking into account the cosmological constant, the nonlinear differential equation that describes the movement of the planets was constructed and exactly solved.

Solving the inverse problem, different theoretical estimates were obtained to calculate the value of the cosmological constant. The obtained results were compared with those of other authors, both theoretical and experimental. In the Schwarzschild cosmological radius $r_{S\Lambda} = 4.1$ km, and the radius limit of the curvature of the gravitational field of the solar system $r_{\text{lim}} = 5.1 \times 10^{10}$ Km $= 340$ UA were obtained, and the existence of several regions was discovered, which were classified into: planetary curvature region, exoplanet curvature region, and the prediction region of the existence of new exoplanets.

The proposed methodology can be of great interest to astronomers, cosmologists, nonlinear physics researchers, and all those interested in the study of the universe.

**Author Contributions:** Conceptualization: J.E.P.Q., J.E.C.H. methodology: J.E.P.Q., J.E.C.H., A.H.S.S. Software: A.H.S.S. Formal analysis: J.E.P.Q., J.E.C.H., A.H.S.S. writing—revision and edition: J.E.P.Q., J.E.C.H., A.H.S.S. All authors have read and agree with the published version.

**Funding:** This research received no external funding.

**Data Availability Statement:** Not applicable.

**Acknowledgments:** The authors thank the University of Francisco Jose de Caldas, University of Tolima, and National university of Colombia for the support to carry out this work.

**Conflicts of Interest:** The authors declare no conflict of interest.

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
