# Peer review of "Calculation of the Cosmological Constant for the Planetary System in Schwarzschild’s Cosmological Model"

_universe, doi:10.3390/universe8090449_

Round 1
Reviewer 1 Report
In this manuscript the author has derived the planetary orbit equations by incorporating the cosmological constant in to GR. Using the planetary perihelion and aphelion data the author have determined the value of cosmological constant using various methods. The manuscript has following issues.
1. The author determines the value of cosmological constant but the confidence intervals of the values also needs to be determined using similar equations.
2. The conclusion and the abstract seems very high resemblance, the conclusion in particular needs to rewritten.
Hence I don't recommend the manuscript for publication in current form.
Author Response
The following changes were made:
1- The abstract and conclusions were written.
2-Confidence intervals were found for each of the tables.
Please see the attached file.

Reviewer 2 Report
In this work, the authors proposed the static cosmological model of the Schwarzschild solution for the solar system, taking into account the cosmological constant in the equation of the General Theory of Relativity (GTR) proposed by A. Einstein. I carefully checked the physical definitions and derivations in the paper. I think the presented analysis and conclusions are reasonable and convincing. Therefore, I would recommend this paper to be published in Universe when the authors address the following issues.
1. In the current manuscript, the Abstract has been divided int so many short paragraphs which is not only unnecessary but also will will hinder the readers from clearly understanding the authors’ ideas. Such comments also applies to the formulation of Conclusions and throughout the paper.
2. In the current manuscript, I can not see the details of all Tables and Figures.
3. In addition to the above concerns, there is problem in the present manuscript. The paper is, however, written not so well in the illustration and discussion of the results. More specifically, in the Section 6 and 7 the authors focus only on the numerical presentation of their results, paying little attention to the quantitative discussion of important systematics. I think there is an extra burden on the authors to do a careful job of considering the systematics fully, especially what quantities and observations will affect the results shown in all Tables and Figures.
4. The authors should thoroughly check the manuscript, concerning many typos exited in the current manuscript.
Author Response
The following changes were made:
1- The abstract and conclusions were rewritten.
2-Confidence intervals were found for each of the tables.
3. Qualitative physical comments were made on section 6 and 7 now section 8.
4. The bibliographical references were expanded.
Thank you so much for the comments.
Please see the attached file

Reviewer 3 Report
Report in the file.

Author Response
The following changes were made:
1- The abstract and conclusions were rewritten.
2-Confidence intervals were found for each of the tables.
3. Qualitative physical comments were made on section 6 and 7 now section 8.
4. The bibliographical references were expanded.
5. Fixed the sign of the cosmological constant in equation 2
6. Equation 8 was corrected.
7. Bibliographic references on metric 14 were added
8. It is clarified that equation 26 is obtained by deriving equation 24
9. Fixed some typographical errors
Thank you so much for the comments.
Please see the attached file
